# Anaerobic Carbon Monoxide Uptake by Microbial Communities in Volcanic Deposits at Different Stages of Successional Development on O-yama Volcano, Miyake-jima, Japan

**DOI:** 10.3390/microorganisms9010012

**Published:** 2020-12-22

**Authors:** Amber N. DePoy, Gary M. King, Hiroyuki Ohta

**Affiliations:** 1Department of Biological Sciences, Louisiana State University, Baton Rouge, LA 70803, USA; adepoy1@lsu.edu; 2College of Agriculture, Ibaraki University, 3-21-1 Chuo, Ami-machi, Ibaraki 300-0393, Japan; hiroyuki.ohta.1494@vc.ibaraki.ac.jp

**Keywords:** carbon monoxide, anaerobic, volcanic, soil, microbial community, diversity, thermophilic

## Abstract

Research on Kilauea and O-yama Volcanoes has shown that microbial communities and their activities undergo major shifts in response to plant colonization and that molybdenum-dependent CO oxidizers (Mo-COX) and their activities vary with vegetation and deposit age. Results reported here reveal that anaerobic CO oxidation attributed to nickel-dependent CO oxidizers (Ni-COX) also occurs in volcanic deposits that encompass different developmental stages. Ni-COX at three distinct sites responded rapidly to anoxia and oxidized CO from initial concentrations of about 10 ppm to sub-atmospheric levels. CO was also actively consumed at initial 25% concentrations and 25 °C, and during incubations at 60 °C; however, uptake under the latter conditions was largely confined to an 800-year-old forested site. Analyses of microbial communities based on 16S rRNA gene sequences in treatments with and without 25% CO incubated at 25 °C or 60 °C revealed distinct responses to temperature and CO among the sites and evidence for enrichment of known and potentially novel Ni-COX. The results collectively show that CO uptake by volcanic deposits occurs under a wide range of conditions; that CO oxidizers in volcanic deposits may be more diverse than previously imagined; and that Ni-dependent CO oxidizers might play previously unsuspected roles in microbial succession.

## 1. Introduction

Microbes colonize fresh volcanic deposits (e.g., lava, cinders, and ash) as soon as temperature, moisture, and nutrient regimes become permissive. However, organic matter, which is initially absent, partially determines the pace of colonization and can severely limit microbial biomass, diversity, and activity long after deposition has occurred [1]. Organic matter availability in turn depends on the development of algal and vascular plant communities, which arise in concert with microbial communities.

In the absence of organic matter, atmospheric trace gases and inorganic reductants can promote survival and even growth of some bacterial populations [1,2]. A variety of organic poor deposits on Kilauea (Hawai’i, USA) and O-yama (Miyake-jima, Japan) volcanoes have been reported to consume molecular hydrogen and CO [1,3]. Uptake rates for both gases were sufficient to account for a significant fraction of overall metabolic activity. Hydrogen and CO uptake also contribute to microbial communities in other organic limited systems, including cold and hot desert soils [4,5].

Under aerobic conditions, a phylogenetically broad group of bacteria oxidize CO to CO_2_ using molybdenum-dependent CO dehydrogenases (Mo-CODH) [6]. Carboxidotrophic CO oxidizers couple high CO concentrations to growth, but many CO oxidizers (termed carboxidovores) use relatively low CO concentrations and are incapable of CO-dependent growth [6]. Mo-dependent CO oxidation occurs ubiquitously in terrestrial and aquatic systems and affects atmospheric CO concentrations by consuming about 10% of the annual atmospheric flux on a global basis [7].

Under anoxic conditions, well characterized nickel-dependent CODHs (Ni-CODH) [8,9,10] oxidize CO to diverse end products, including CO_2_ + H_2_ (hydrogenogenesis), acetate (acetogenesis), and methane + CO_2_ (methanogenesis). Sulfidogens and some iron-reducing bacteria can also oxidize CO to CO_2_ using Ni-CODH [10], while the purple non-sulfur bacteria can use CO as an electron donor during anoxygenic photosynthesis [11]. In all of these cases, high CO concentrations can support growth.

Although they are metabolically diverse, Ni-dependent CO oxidizers (Ni-COX) have been primarily cultivated from the thermophilic Firmicutes [12] with the first novel thermophilic Crenarchaeote reported only recently [13]. A metagenomic, stable isotope probing analysis of several hot springs (45 °C–65 °C) confirmed Ni-COX dominance by Firmicutes, but also revealed Betaproteobacteria affiliated with *Azonexus* in a spring at 45 °C [14]. Mesophilic examples otherwise include a few Proteobacterial anoxygenic phototrophs [15,16] and a small number of clostridia, among them *Clostridium difficile* [17]. These observations suggest that Ni-COX distributions and activity in situ might be constrained primarily to geothermally-heated aquatic and terrestrial systems. However, genome surveys have revealed Ni-CODH in numerous phyla and classes not previously known for carboxidotrophy [18], so the diversity and distribution of Ni-COX might prove far greater than previously imagined.

Regardless, little is known about the potential activity of Ni-COX or their impacts on CO cycling. Some insights might be inferred from existing knowledge of acetogens, methanogens, and sulfidogens, but these studies typically do not address CO transformations specifically [19]. Results from studies of hydrogenogenic CO oxidation also offer useful insights, but they are basically confined to extreme terrestrial and marine systems [20,21,22,23,24].

We report here results from a comparative analysis of aerobic (Mo-dependent) and anaerobic (Ni-dependent) CO uptake with 10 ppm CO, and Ni-dependent CO uptake at ambient temperature and 60 °C with 25% CO. CO uptake assays were conducted using samples from O-yama Volcano on Miyake-jima Island (Japan); one site was comprised of a mature forest on an approximately 800-year-old volcanic deposit; two additional sites were impacted by a recent eruption (July–August 2000). Plant colonization on one of the younger sites was extensive while colonization of the other was sparse and patchy. Results enabled tests of hypotheses that: (1) Mo-dependent CO uptake capacity exceeds that for Ni-dependent CO uptake at low CO concentrations; (2) thermophilic CO uptake capacity exceeds that for mesophiles; (3) CO uptake rates at 10 ppm and 25% concentrations increase with organic matter and plant development in volcanic deposits. The second and third hypotheses were confirmed, while the first was not. In addition, we described microbial community composition across all sites (derived from 16S rRNA gene sequences) and responses of communities to both temperature shifts and elevated CO. Community responses to both variables were distinct for each of the sites and revealed potentially novel Ni-COX diversity.

## 2. Materials and Methods

### 2.1. Site Descriptions, Sample Collection and Processing

Three previously described sites were sampled on O-yama Volcano (Miyake-jima, Japan): OY, IG-7 and CL [3,25,26]. Briefly, O-yama erupted explosively in July–August 2000, resulting in extensive ash and tephra deposits and SO_2_ emissions [26,27]. Sites OY (34.0802 N, 139.5190 E) and IG-7 (34.0894 N, 139.5141 E) were located near the volcano summit (about 600 m and 540 m, respectively) and were vegetated prior to the eruption; both were heavily impacted by ash fall with a loss of vegetation cover; OY was more heavily affected by SO_2_ emissions than IG-7. At the time of sampling, OY supported patchy vegetation, while IG-7 supported dense stands of the grass, *Miscanthus condensatus*. CL (34.1119 N, 139.5015 E; approximately 120 m) was relatively unaffected by the eruption, and was dominated by Itajii chinkapin, *Castanopsis sieboldii*, in a climax forest on lava flows >800 years old. Triplicate samples from the upper 5 cm depth interval from all sites were collected with a small ethanol-sterilized spatula in March 2019. Samples were stored in Ziplock bags at ambient temperature and transferred to a laboratory at Louisiana State University where they were held at ambient temperature for further processing (about 1 week).

### 2.2. CO Uptake Assays

Replicate samples from each site were used to create four sets of treatments, each with triplicates. All treatments were initiated by transferring 5 g fresh weight (gfw) samples to 60 mL serum bottles that were sealed with blue butyl rubber stoppers. Two treatments involved incubations with aerobic or anaerobic headspaces containing approximately 10 ppm CO added from 1000 ppm oxygen-free or air stocks as appropriate. Anaerobic headspaces were established by flushing with deoxygenated nitrogen, while aerobic headspaces were comprised of ambient air. After adding CO, the serum bottles were incubated statically at ambient temperature, and the bottle headspaces were sub-sampled at suitable intervals with air- or nitrogen-flushed needles and syringes as appropriate. CO concentrations were analyzed using a Peak Laboratories (Mt. View, CA, USA) Peak Performer 1 equipped with a reduced gas detector and a Molecular Sieve 5A column (1 m × 6.25 mm outside diameter (OD) stainless steel) operated at 120 °C with an ultrahigh purity air carrier gas. CO uptake was followed until concentrations reached ambient or sub-ambient levels. Uptake rates were determined by linear regression analysis of concentration changes during the early phase of incubation. CO uptake rates were expressed on a dry weight basis.

Two separate trials with two additional sets of triplicates from each site were established using 60 mL serum bottles, the headspaces of which were flushed with deoxygenated nitrogen. CO was added from a 100% stock to a final concentration of 25%. One set of bottles for each site was incubated statically at ambient temperature while the other set was incubated statically at 60 °C. Headspaces were sampled at intervals with a nitrogen-flushed needle and syringe for CO analysis using an SGI 8610C Gas Chromatograph (Folsom, CA, USA) equipped with a thermal conductivity detector and a Molecular Sieve 5A column (2 m × 6.25 mm OD stainless steel) operated at 60 °C. Maximum CO uptake rates were estimated from linear regression analyses of CO concentrations or in some cases from the products of first-order uptake rate constants and initial CO concentrations. These analyses excluded periods during which there were initial lags in uptake.

Since some replicates in both trials from IG-7 and OY did not consume 25% CO under some conditions while all replicates oxidized CO at 10 ppm, a separate analysis was used to assess the possibility of inhibition at elevated CO concentrations. Sets of triplicates were prepared as above for IG-7 and OY. OY headspace CO was adjusted to concentrations of 1%, 5%, 15%, or 25%, and sets of triplicates at each concentration were incubated at 25 °C or 60 °C. Sets of triplicates from IG-7 were incubated at 25 °C with headspace CO concentrations of 0.1%, 1%, 5%, 15%, and 25% CO. CO concentrations were assayed at intervals as described above.

### 2.3. Soil Analyses

Water contents were determined after drying soil sub-samples in an oven at 80 °C for >48 h. Organic matter contents were estimated using “mass loss on ignition” of oven-dried samples combusted at 550 °C for 3 h in a muffle furnace [28,29]. Sample pH values were measured using slurries with a 1:2 ratio of soil and deionized water and a Beckman ion analyzer. All assays were conducted in triplicate for each site.

### 2.4. DNA Extraction, Sequencing and Analysis

Responses of microbial communities to CO additions and elevated temperature were assessed using the four sets of triplicate samples from each of sites CL, IG-7, and OY that were used in the second trial analysis of CO uptake rates. Two sets from each site were incubated at ambient temperature with or without 25% CO as described previously, and two additional sets were incubated with or without 25% CO at 60 °C. CO concentrations were monitored as before. After terminating the CO uptake assays, soil sub-samples were collected from each of the replicates and treatments and stored at −80 °C until they were extracted using a DNeasy PowerSoil extraction kit (Qiagen Inc., Hilden, Germany) following the manufacturer’s instructions. Extracted DNA was visualized by gel electrophoresis and then shipped on dry ice to the Research Technology Support Facility at Michigan State University for multiplexed sequencing of the V4–V5 region of the 16S rRNA genes using primers 515f and 806r with an Illumina Miseq platform (San Diego, CA, USA) with a 2 × 250 bp paired end chemistry [30].

Sequences were processed with the DADA2 pipeline [31]. The SILVA v 132 database was used for classification [32]. Sequencing yielded a total of 2,689,759 reads, which have been deposited with the NCBI SRA as PRJNA673894. After processing sequences through the DADA2 pipeline there were 1,944,382 total reads. One sample, “OY_No_CO 60 °C_2”, had only 28 reads and was removed from any further analyses. After removal of this sample, reads varied from 444 to 128,803. Analyses and visualizations were conducted using the phyloseq R package [33]. Prior to analysis, the most informative amplicon sequence variants (ASVs) were selected based on abundance and variance (at least four counts in 10% of the samples). To preserve the information in the samples with <2000 reads for beta diversity and taxonomic analyses these samples were separated before filtering. For alpha diversity estimates (Chao1 (richness) and the Shannon Index (abundance and evenness)), the data were rarefied to the minimum sample size. This rarefaction removed reads <10,609. For beta diversity analyses, two different data transformations were used with filtered data, a centered log ratio transformation (CLR) [34] and a Hellinger transformation [35]. Both sets of transformed data were ordinated using Redundancy analysis (RDA).

### 2.5. Statistical Analyses

Sites that had at least one replicate with CO uptake were included in the statistical analysis. Replicates that had no CO uptake at 25% CO were assigned a zero rate. Therefore, CO uptake rates for 25% concentrations were transformed prior to statistical analyses using a log(1 + x) method, while uptake rates for 10 ppm CO were transformed with log(x). For 10 ppm CO treatments, uptake rates among sites were compared using a one-way ANOVA for each treatment, oxic and anoxic. To assess differences between treatments at each site, uptake rates were compared using a two-way ANOVA with an interaction between site and treatment. Similarly, 25% CO treatment uptake rates among sites were compared using a one-way ANOVA for each temperature, 25 °C and 60 °C. For taxonomic results, a Student’s *t*-test was run for genera that varied markedly between no exogenous CO and 25% CO treatments. For beta diversity metrics, a PERMANOVA was run on each distance matrix (Hellinger and Aitchison) with interactions between site, temperature, and CO.

## 3. Results

### 3.1. CO Uptake

CO uptake rates at 10 ppm were used to compare potential oxic (Mo-CODH) and anoxic (Ni-CODH) activities. At all sites and for both treatments, CO uptake occurred with no observable lag. Anoxic uptake rates varied from 0.6 nmol gdw^−1^ d^−1^ at OY to 39.8 nmol gdw^−1^ d^−1^ at CL under anoxic conditions. Uptake rates differed significantly among sites (*p*_oxic_ = 1.46 × 10^−5^, *p*_anoxic_ = 0.00018), but did not differ significantly between oxic and anoxic treatments (*p* = 0.373). For all sites and both treatments, CO approached lower limits of detection with values well below ambient atmospheric levels.

Results for 25% CO concentrations were more variable. For CL, all replicates at each temperature oxidized CO in two separate trials. However, for IG-7, 2 of 3 and 3 of 3 replicates oxidized CO at 25 °C and 60 °C, respectively, in one trial, while 0 of 3 and 2 of 3 replicates were active at 25 °C and 60 °C, respectively, in a second trial. For OY, no activity was observed at 25 °C in either of two trials, while 3 of 3 and 2 of 3 replicates were active at 60 °C in separate trials.

CO uptake rates in the first trial differed significantly among sites compared at each incubation temperature (ambient and 60 °C; *p* = 0.0026 and 0.0001, respectively). Excluding replicates with no activity at 25 °C, uptake rates ranged from 5.7 µmol CO gdw^−1^ d^−1^ to 26.8 µmol CO gdw^−1^ d^−1^ at sites IG-7 and CL, respectively (Figure 1, Table 1). CO uptake rates for 60 °C incubations ranged from 4.2 µmol CO gdw^−1^ d^−1^ to 166.2 µmol CO gdw^−1^ d^−1^ at sites OY and CL, respectively (Figure 1, Table 1). For each site, CO uptake rates at 60 °C were significantly greater than at 25 °C (*p* = 0.0002). CO uptake rates for OY were similar for the first and second trials; rates for the two trials were also similar for CL at 25 °C but lower at 60 °C for the second trial (Table 1). For IG-7, rates were lower for the second trial at both temperatures. Overall, trends for Trial 1 and Trial 2 were comparable.

CO uptake for 25% was less reproducible for IG-7 and OY replicates, so a survey of uptake rates at lower concentrations was conducted to address the possibility of CO sensitivity. The results showed that activity for IG-7 and OY replicates was more reproducible with CO concentrations <25%. All IG-7 25 °C replicates showed uptake at 0.1% and 1%, but no activity at concentrations ≥5%. All OY replicates were active at 1% CO for 25 °C and 60 °C incubations, but no activity was observed at higher concentrations. Rates for 0.1% CO varied from 0.07 ± 0.002 µmol CO gdw^−1^ d^−1^ to 0.21 ± 0.06 µmol CO gdw^−1^ d^−1^ (Table 2). OY 60 °C and IG-7 25 °C uptake rates at 1% CO concentrations were 0.37 ± 0.02 and 0.34 ± 0.01 µmol CO gdw^−1^ d^−1^, respectively.

Apparent lag times were defined as the time in days before a decline in CO was observed. Apparent lag times for 25 °C incubations ranged from 9.9 d to 19.1 d for CL and IG-7, respectively, while for 60 °C apparent lag times ranged from 0.9 d for CL to 1.8 d and 11.7 d for IG-7 and OY, respectively (Figure 2). Apparent lag times were significantly greater at 25 °C for both CL and IG-7 (*p* = 0.006) (Figure 2). Apparent lag times also varied across sites (CL < IG-7 < OY) and increased in parallel with increasing levels of exogenous CO. Apparent lag times for 0.1% CO concentrations were consistently <1 day, while apparent lags for 1% concentrations varied from 2.1–9.9 days.

### 3.2. Microbial Community Analysis

For CL and IG-7, the sites with robust plant development, microbial communities at 25 °C without exogenous CO were dominated by eight phyla present at relative abundances >1%, collectively accounting for 90–99% of all taxa (Appendix A, Figure 3A). Proteobacteria, Acidobacteria, Verrucomicrobia, Bacteroidetes, and Planctomyces were the most prominent phyla at both sites declining in the order presented. A similar pattern was observed for OY with the exceptions that Proteobacteria and Verrucomicrobia were lower in abundance and Chloroflexi were considerably higher than at the other sites. Indeed, Chloroflexi appeared to decline in abundance from the least developed site to the most mature (CL) with intermediate values at IG-7 (Figure 3A, Appendix A). OY also supported several phyla that were rare or absent from IG-7 and CL, including Thaumarchaeota, Elusimicrobia, Patescibacteria, and Candidate Phyla FCPU426 and WPS-2.

Genera varied among sites for samples incubated at 25 °C without added CO. Sixteen genera were present at CL in relative abundances >1%, accounting for 60.3% of the total (Figure 3B, Appendix A). Most of these genera were present at low abundances with the exception of *Candidatus* Udaeobacter, which accounted for 14.0 ± 0.63% of the total. For IG-7, the most abundant genera were *Brayobacter* (6.46 ± 1.16%) and *Burkholderia-Caballeronia-Paraburkholderia* (5.96 ± 0.70%); for OY, two genera were present at relative abundances >5%, *Nevskia* and *Phenylobacterium*. The other genera present at >1% collectively accounted for 49.3% of all taxa.

Ten phyla accounting for 97.8% of all taxa were observed for CL communities at 25 °C with 25% CO. The most abundant phyla were Proteobacteria and Acidobacteria (Figure 3A, Appendix A). In contrast to CL, samples from IG-7 and OY did not oxidize CO during trials used for community analysis. Nonetheless, Firmicutes dominated IG-7 (33.1 ± 6.68%), while Proteobacteria and Chloroflexi (32.8 ± 5.09% and 23.6 ± 1.25%, respectively) dominated OY (Figure 3A, Appendix A).

Genera also varied among sites for samples incubated at 25 °C with CO. Fifteen genera present at relative abundance >1% accounted for 60.6% of all CL taxa with *Candidatus* Udaeobacter the most abundant (15.5 ± 0.60%). Three other genera were present at abundances >5%, *Pseudolabrys, Candidatus* Xiphinematobacter, and *Bradyrhizobium,* (Figure 3B, Appendix A). *Desulfitobacterium* and *Clostridium sensu stricto* group 12 dominated IG-7 with five additional genera present at relative abundances >1% (Figure 3B, Appendix A). Seventeen genera present at relative abundances >1% accounted for 69.9% of all taxa at OY. *Sediminibacterium* and *Opitutus* were the most abundant (25.4 ± 12.0 and 9.58 ± 3.57%, respectively).

For sites IG-7 and OY, communities at 60 °C without exogenous CO were dominated by Firmicutes (97.9 ± 0.84% and 95 ± 3.30%, respectively); Proteobacteria were substantially reduced relative to compositions at 25 °C without CO (0.14 ± 0.10% and 3.22 ± 3% for IG-7 and OY, respectively). In contrast, CL communities were dominated by eight phyla present at relative abundances >1%, collectively accounting for 96% of all taxa, with two of the most dominant phyla, Firmicutes and Proteobacteria, accounting for 78% of the total (Figure 3A, Appendix A).

At a genus level, the composition of communities at 60 °C without added CO also varied among sites. For CL, 11 genera were present at relative abundances >1%. In decreasing order, *Tumebacillus, Paenibacillus, Bacillus,* and *Thermoanaerobacterium* were the most abundant. For IG-7, five genera with relative abundances >1% accounted for 96.8% of all taxa, including *Thermoanaerobacterium* at 42.8 ± 25.1%. For OY, four genera with relative abundances >1% accounted for 50.7% of all taxa, including *Krypidia*, *Thermoanaerobacterium*, *Tumebacillus*, and *Brevibacillus* (Figure 3B, Appendix A).

For site CL, seven phyla representing 94.8% of all taxa were present at relative abundances >1% for communities incubated with 25% CO at 60 °C. Firmicutes (52.8 ± 4.17%) and Proteobacteria (28.1 ± 1.59%) were the most abundant (Figure 3A, Appendix A). IG-7 and OY were dominated by Firmicutes (97.3 ± 1.09% and 99.7 ± 0.30%, respectively).

The dominant genera varied among sites for communities incubated at 60 °C with CO. Thirteen genera present at relative abundances >1% accounted for 77.4% of the taxa at CL. Five of these genera were present at relative abundances >5%: *Geobacillus*, *Paenibacillus*, *Bacillus*, *Tumebacillus*, and *Tuberibacillus* (Figure 3B, Appendix A). For IG-7, two replicates oxidized CO uptake while a third did not. The CO-oxidizing replicates contained three abundant genera, *Tumebacillus*, *Moorella*, and *Bacillus*; in contrast, nine genera present at >1% occurred in the replicate with no activity, including *Thermoanaerobacterium*, *Bacillus*, *Effusibacillus*, *Alicyclobacillus*, and *Sulfobacillus* (Figure 3B, Appendix A). For site OY, replicate 1 exhibited CO uptake, while two did not. The replicate that exhibited CO uptake had four abundant genera, Peptococcaceae SCADC1-2-3, *Moorella*, *Tumebacillus*, and *Thermoanaerobacterium*. The replicates that did not oxidize CO had four abundant genera, Peptococcaceae SCADC1-2-3, *Moorella*, *Gelria*, and *Caldanaerobius* (Figure 3B, Appendix A).

Alpha diversity indices (Chao1 and Shannon Index) were estimated using phyloseq after rarefying to the minimum sample read depth (10,609). Chao1 estimates for species richness (Figure 4, Appendix A) varied from 82.6 (range 49–116.2) for IG-7 with 25% CO at 60 °C to 1728.5 ± 25.0 for CL with no CO at 25 °C. In general, values declined from CL to IG-7 to OY and were lower at 60 °C than at 25 °C. The Shannon Index (Figure 5, Appendix A) ranged from 1.603 ± 0.348 (IG-7 with no CO at 60 °C) to 6.693 ± 0.019 (CL with no CO at 25 °C). The Shannon Index generally followed trends for Chao1, with values declining from CL to OY and with elevated temperature. Incubation with CO did not have a statistically significant effect on either index.

Beta diversity indices were used to assess relationships among sites, temperature, and CO treatments. Ordinations based on CLR transformations (Aitchison distance) and the Hellinger distance were performed on filtered data with the most informative ASVs (4234 ASVs). Both distance metrics yielded similar results. There were no significant differences between T0 controls and communities from samples incubated at 25 °C (Figure 6), but these communities differed significantly among the three sites (*p* = 0.001). In contrast, communities from IG-7 and OY samples incubated at 60 °C were indistinct but differed significantly from CL communities at 60 °C. For each site, communities differed significantly between 25 °C and 60 °C (*p* = 0.001) but CO additions had no clear effect overall (*p* = 0.201).

## 4. Discussion

CO uptake attributed to Mo-dependent CO dehydrogenases has been observed for a wide range of physiologically and ecologically versatile Bacteria and some Archaea [6,36] in freshwater, marine, and terrestrial systems, including volcanic deposits and deserts [1,3,5,37,38,39]. CO uptake at low concentrations by soils in particular contributes significantly to the global atmospheric CO budget [7]. In contrast, CO uptake attributed to Ni-dependent CO dehydrogenases has been reported for a phylogenetically more restricted group of Bacteria and Archaea, most of which are thermophilic [10,18,40,41]. The molecular biology, biochemistry, and physiology of selected isolates have been characterized extensively [9,10,15,18,42], but the ecological significance of Ni-COX remains largely unknown with a few exceptions.

Extensive studies of hot springs have resulted in isolation of numerous thermophilic, hydrogenogenic Ni-COX and provided insights about their distribution [21,24,43,44]. Applications of stable and radioisotopic probing approaches and CO uptake assays have revealed a relatively low diversity for active Ni-COX, but also indicated that CO uptake can contribute significantly to hot spring community metabolism when CO is relatively abundant [14,20,45]. Other studies have addressed Ni-COX in sewage sludge [46,47], the use of Ni-COX in microbial fuel cells [48,49] and the prospects for bioconversions of syngas to hydrogen [41].

Research on Ni-COX activity in other systems has been much more limited. Inman et al. [50] found no anaerobic CO uptake by soils, in contrast to observations by Bartholomew and Alexander [51] and Conrad and Seiler [52]. The latter studies observed anaerobic uptake with no lag at low CO concentrations, but the source of activity was unknown. King [39] compared CO uptake by forest soils from Maine, Georgia and Hawai’i (USA) at concentrations of 1–10 ppm under oxic and anoxic conditions. Uptake rates were somewhat higher under oxic than anoxic conditions, but anoxic soils from all sites reduced CO concentrations below atmospheric levels. King [39] further showed that nitrate additions had no effect on aerobic or anaerobic CO uptake while chloroform inhibited anaerobic but not aerobic uptake. These results indicated that anaerobic CO uptake was likely due to Ni-COX activity. In a separate study, King [53] documented anaerobic CO uptake by surface and sub-surface sediments from a Maine salt marsh. The observed activity was attributed to Ni-COX, based on the outcomes of studies with inhibitors and nitrogen oxides [53].

In the study reported here, Mo-dependent (aerobic) and Ni-dependent (anaerobic) CO uptake rates at 10 ppm were highest at CL, a forested site with the highest organic content, while rates were similar for the two recent volcanic deposits (IG-7 and OY, Table 1). This observation is consistent with previous reports, which posited that higher organic matter supports larger microbial communities with greater CO uptake capacity [38,54]. In addition, aerobic and anaerobic rates were comparable at each of the sites, which contrasts to some extent with previously reported trends for mature continental forest soils [39,52]. Regardless, the data collectively indicate the capacity for Ni-COX activity at atmospheric CO concentrations is widespread in soils, occurring even in recently formed soils (IG-7 and OY). At each of the sites aerobic and anaerobic CO uptake occurred without a lag, and in both cases CO concentrations were reduced to sub-atmospheric levels. The absence of a lag in this and other studies [39,51,52] indicates that Ni-COX are either activated very rapidly after anoxic conditions are established, or that they include oxygen tolerant populations that can maintain limited activity under conditions that prevail in situ. The fact that acetogens in soil have been described as aerotolerant [19,55] suggests that both possibilities are likely. The ability of Ni-COX to consume CO at sub-atmospheric levels suggests that they might play underappreciated roles in the atmospheric CO budget and in CO cycling, particularly in systems that experience intermittent anoxia or sub-oxic conditions.

At present the Ni-COX populations responsible for anaerobic CO uptake at ambient concentrations remain a matter of speculation. Inventories of communities in time zero samples reveal several putative Ni-COX taxa, including *Clostridium*, *Desulfitobacterium*, *Geobacillus*, and *Geobacter*, all at low abundance (Figure 3B). However, the capacity of known Ni-COX to oxidize CO at atmospheric levels has not yet been confirmed. Further, and in contrast to possibilities for Mo-COX, the toolkit for identifying Ni-COX populations active at low CO concentrations remains limited by the extensive sequence diversity of the Ni-dependent CO dehydrogenase and by the extremely low levels of carbon that can be incorporated into biomass from atmospheric CO.

Incubations with high CO concentrations sacrifice potential insights about uptake in situ but offer options to address other questions. In this study, incubations with 25% CO headspaces were used to address questions about differential responses to elevated CO as a function of temperature, the distribution of putative carboxidotrophic Ni-COX across an ecosystem gradient, and the identity of putative Ni-COX active at 25 °C and 60 °C. The latter questions were enabled by the use of substrate-level CO additions. The amount of CO added to each of the samples, about 700 µmol, was sufficient to support cell growth equivalent to approximately 830 µg of biomass and 1.7 × 10^−9^ cells, assuming a 5% growth efficiency, complete utilization of the added CO, cell carbon contents of 50% and a mass per cell of 5 × 10^−13^ g. With sample masses of 5 g fresh weight, carboxidotrophic Ni-COX growth could have produced approximately 3.4 × 10^8^ cells gfw^−1^. This likely represented a small to modest fraction of total cell abundance that could have led to detectable changes in community compositions during incubations with and without added CO.

However, in contrast to consistent activity for all sites and replicates at 10 ppm, uptake of 25% CO was variable. All replicates for both incubation temperatures at CL completely consumed the added CO in two separate trials, but for IG-7, 2 of 6 replicates in two trials at 25 °C and 5 of 6 replicates at 60 °C oxidized CO. For OY, 0 of 6 and 5 of 6 replicates were active in two trials at 25 °C and 60 °C, respectively. Although reasons for the different outcomes of IG-7 and OY trials are unclear, evidence suggests that Ni-COX populations at IG-7 and OY might be inhibited by CO concentrations ≥1%, especially at 25 °C. In support of this hypothesis, CO uptake at 25 °C was observed for OY with 1000 ppm CO and IG-7 with 1% CO, but not at higher concentrations (Table 2). These results suggest that at some sites mesophilic Ni-COX populations might be more sensitive to CO than thermophiles. However, the extent to which CO sensitivity reflects successional status or other variables is unknown.

Although a detailed analysis of temperature responses was not conducted in this study, apparent lag times were shorter and the capacity for CO uptake was greater at 60 °C than at 25 °C across all sites, suggesting a thermophilic optimum. This might have reflected differences in CO sensitivity of mesophiles and thermophiles or other factors. For example, thermophilic Ni-COX Firmicutes spores might have occurred at relatively high densities that could have led to larger populations than those of Ni-COX mesophiles and thus resulted in higher uptake rates.

Trends in apparent lag times support this interpretation. Because low CO uptake rates at the initiation of the incubations were not detectable due to limitations of the analytical system, the observed lags reflected the time required for populations to increase their uptake capacity to a degree that changes in headspace CO could be observed. That could have resulted from increased elaboration of Ni-CODH in static populations, spore germination and population growth, or both. However, irrespective of the mechanism involved, thermophilic Ni-COX clearly responded more rapidly than their mesophilic counterparts, which is remarkable since thermophiles in temperate soils rarely experience permissive growth temperatures while mesophiles routinely do.

Since CO was added at substrate-level concentrations, analysis of 16S rRNA genes in soils with and without CO could reveal Ni-COX populations that were able to grow during the assays and their distributions among sites (see Figure 7). In this study, interpretations are constrained by inconsistent CO uptake for IG-7 and OY at 25 °C and 60 °C, and low DNA yields for OY at 60 °C. Nonetheless, several insights are possible. Inspection of the composition of CL communities incubated at 60 °C (Appendix A, Figure 3B) revealed enrichment of *Geobacillus*, a genus known to harbor Ni-COX [56]. In addition, *Brevibacillus* and *Tuberibacillus* were enriched in the presence of CO. Although Ni-CODH has not been reported for either genus, both might harbor as yet uncharacterized strains with a capacity for anaerobic CO uptake. In contrast, no obvious Ni-COX enrichments were evident for CL samples incubated with CO at 25 °C (Appendix A, Figure 3B), even though they contained *Geobacillus*. The lack of enrichment in these samples might reflect uptake by multiple taxa resulting in only small changes for each.

For IG-7 at 60 °C, *Moorella*, a genus known to harbor CO oxidizers (e.g., Fukuyama et al. [42]), was enriched in the replicate that oxidized CO, but it was not enriched in the remaining replicates (Appendix A, Figure 3B). *Tumebacillus* was also abundant in the replicate that oxidized CO, but it has not yet been documented as a CO oxidizer. None of the IG-7 replicates at 25 °C oxidized CO, but the replicates containing CO differed in composition from those that did not (Appendix A, Figure 3B) and were enriched in genera known to oxidize CO or to contain Ni-CODH, e.g., *Desulfitobacterium* and *Clostridium* [15,57]. This suggests that exogenous CO altered IG-7 communities even when CO was not metabolized.

Like IG-7, *Tumebacillus* and *Moorella* were enriched in the OY replicate at 60 °C that oxidized CO (Appendix A, Figure 3B). This replicate was also dominated by a Peptococcaceae ASV that did not occur in the absence of CO and that might be a novel CO oxidizer. In contrast to IG-7, OY replicates with and without CO at 25 °C were not strongly differentiated and Ni-COX genera were not abundant (Appendix A, Figure 3B). At this site, CO did not appear to affect communities when it was not oxidized. The difference between IG-7 and OY in this context might be an outcome of basic differences in community composition ab initio.

Redundancy analysis based on Hellinger and CLR transformations indicated that communities from all sites incubated at 25 °C differed distinctly from those incubated at 60 °C due to losses of mesophiles and gains of thermophiles that can be expected for a substantial temperature upshift (Figure 6). Both metrics also showed that communities from 25 °C differed among the three sites, while those from 60 °C were similar for IG-7 and OY but distinct for CL (Figure 6). The former results along with the distinction of CL at 60 °C undoubtedly reflect differences in age and successional development (Table 1). For IG-7 and OY, community differences for time zero and 25 °C samples appear to reflect the dense *M*. *condensatus* cover at IG-7 and patchy vegetation at OY. The similarity of samples from these sites when incubated at 60 °C suggests that they had accumulated equivalent thermophilic taxa, which might be anticipated given their comparable ages and proximity, but that there had been insufficient time for divergence resulting from differences in plant cover and other ecological variables at the two sites.

Trends in alpha diversity metrics support interpretations of beta diversity analyses. Specifically, Chao1 estimates of richness (Figure 4) distinguish the three sites, with a decline in richness from the oldest, forested site (CL) to the youngest and least vegetated site (OY). Differences between CL and IG-7 suggest that both deposit age and the extent of plant community development affect recruitment and retention of soil microbes on O-yama; differences between IG-7 and OY, which are the same age, suggest that during early succession plant colonization plays a primary role, since both sites presumably experience comparable inputs of potential pioneering taxa due to their proximity.

Although CO did not appreciably affect richness at 25 °C or 60 °C, substantial decreases in richness at 60 °C for IG-7 relative to CL suggest that deposit age and plant development affect O-yama thermophiles as well as mesophiles (Figure 4). Since the former are likely rarely active, differences in richness between the two sites presumably reflect differences in the net accumulation of taxa in microbial seed banks. Additional comparative analyses with a greater range of deposit ages and plant development will be necessary to partition the variables that affect thermophile recruitment. An existing conceptual model invokes atmospheric deposition of primarily *Geobacillus* and then local factors that determine loss rates [58]. However, this model does not account for the variety of taxa observed in this study or specific variables important at local scales.

The Shannon Index offers similar insights to those from Chao1 with some exceptions (Figure 5). Similarities for CL and IG-7 but lower values for OY suggest that plant cover structures community diversity to a greater degree for these sites than deposit age. This likely arises from impacts of aboveground biomass and roots as sources of organic matter and the latter as an agent for generating spatial structure. Decreases in the Shannon Index at 25 °C for IG-7 and OY but not CL after CO addition (Figure 5) suggest that communities with lower richness might be more susceptible to disturbances than communities with higher richness. However, the absence of a CO effect for samples incubated at 60 °C indicates that temperature was a more important structuring agent than CO for thermophiles. Nonetheless, diversity decreased much more dramatically for IG-7 than CL, which suggests that richness remains a factor in disturbance responses at elevated temperatures.

In summary, this study provides evidence for potential Ni-CODH activity in mesothermic soils at low (10 ppm) and high (25%) CO concentrations, including soils formed recently after a volcanic eruption in 2000. These results expand insights from earlier studies of mature continental soils and also reveal populations of thermophilic Ni-COX in young and mature soils capable of consuming high CO concentrations at 60 °C. Analyses of 16S rRNA genes in soils with and without added CO at 25 °C and 60 °C illustrate general trends in microbial community succession across sites that vary in vegetational development while providing indications of the possible identity of some CO oxidizers. The latter include known CO-oxidizing taxa, such as *Geobacillus* and *Moorella*, and potentially novel taxa in the genera *Brevibacillus* and *Tuberibacillus*. Additional assays of other terrestrial and aquatic environments will help clarify the distribution and controls of Ni-COX and their possible contributions to CO cycling.

## Figures and Tables

**Figure 1 microorganisms-09-00012-f001:**
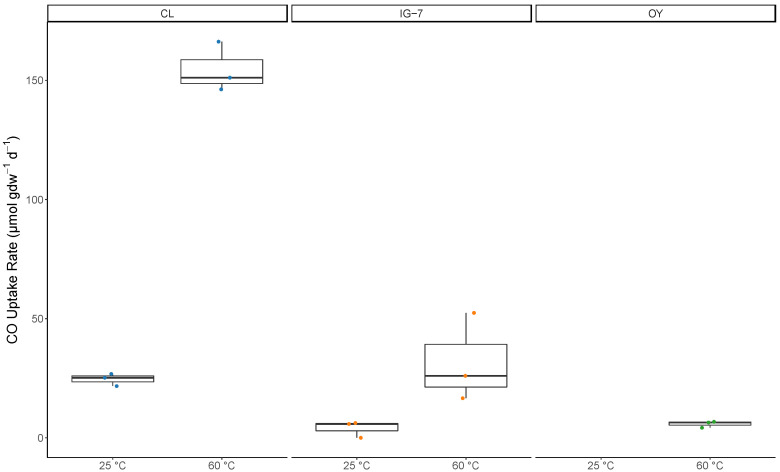
Box and whisker plot representation of CO uptake rates (µmol gdw^−1^ d^−1^) for each temperature (25 °C and 60 °C) at each site (CL, IG-7, and OY) with 25% CO. Median values are indicated by solid bars. Data not present (OY 25°C) indicate no CO uptake.

**Figure 2 microorganisms-09-00012-f002:**
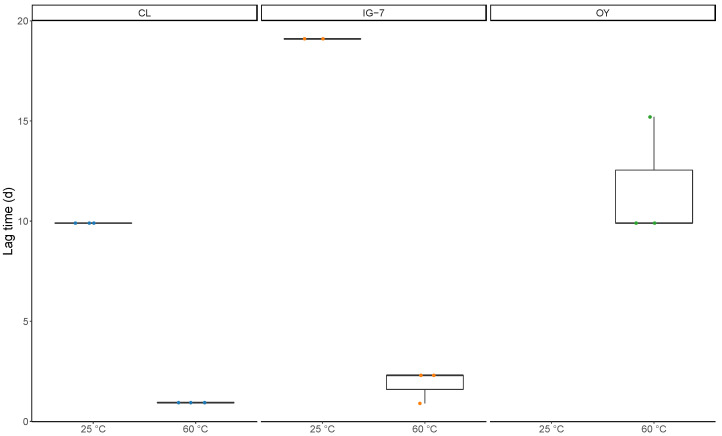
Box and whisker plot representation of apparent lag times (d) for each temperature (25 °C and 60 °C) at each site (CL, IG-7, and OY). Lag times are from CO uptake assay (Trial 1) with 25% CO. Median values are indicated by solid bars. Data not present (OY 25 °C) indicate no CO uptake.

**Figure 3 microorganisms-09-00012-f003:**
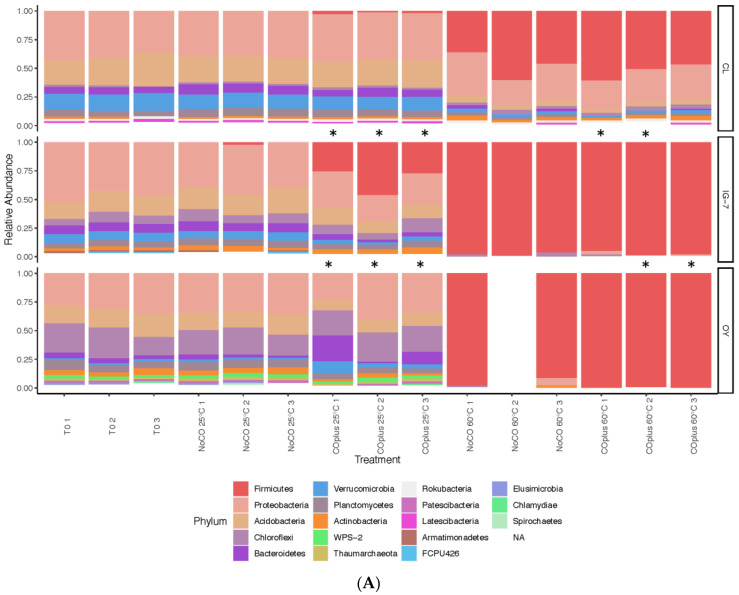
Taxonomic composition for sites CL, IG-7, and OY, at each treatment for T0, No CO 25 °C, 25% CO 25 °C, No CO 60 °C and 25% CO 60 °C. (**A**) Phyla represented are present at relative abundances >1%; (**B**) Genera are present at relative abundances >2%. * Indicates replicates that had no CO uptake activity for Trial 2.

**Figure 4 microorganisms-09-00012-f004:**
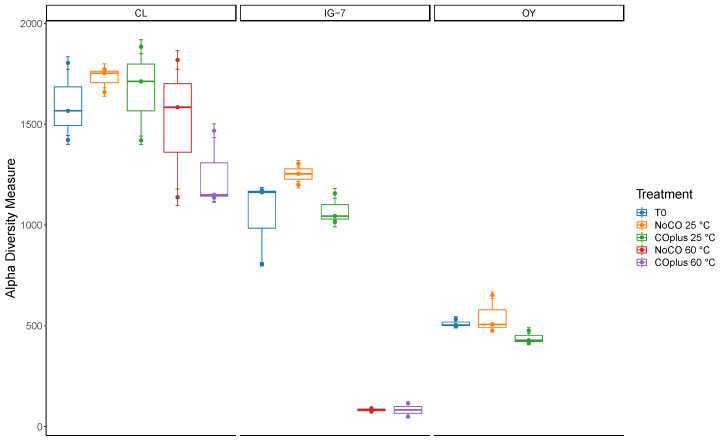
Chao1 index for all treatments for sites CL, IG-7, and OY. Bars in the box and whisker plots represent median values.

**Figure 5 microorganisms-09-00012-f005:**
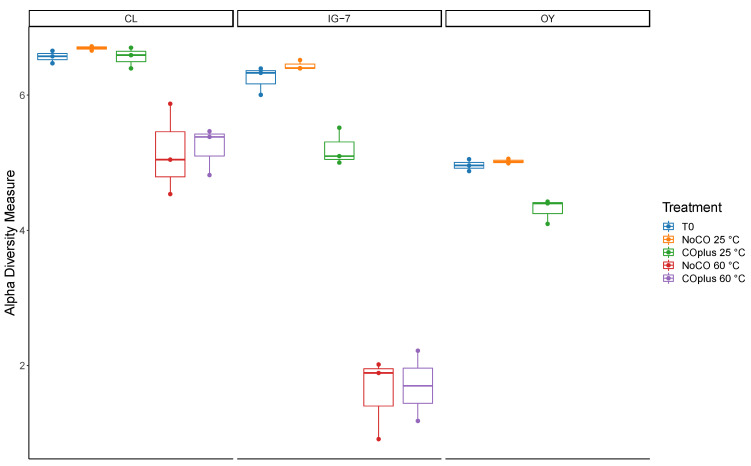
Shannon index for all treatments for sites CL, IG-7, and OY. Bars in the box and whisker plots represent median values.

**Figure 6 microorganisms-09-00012-f006:**
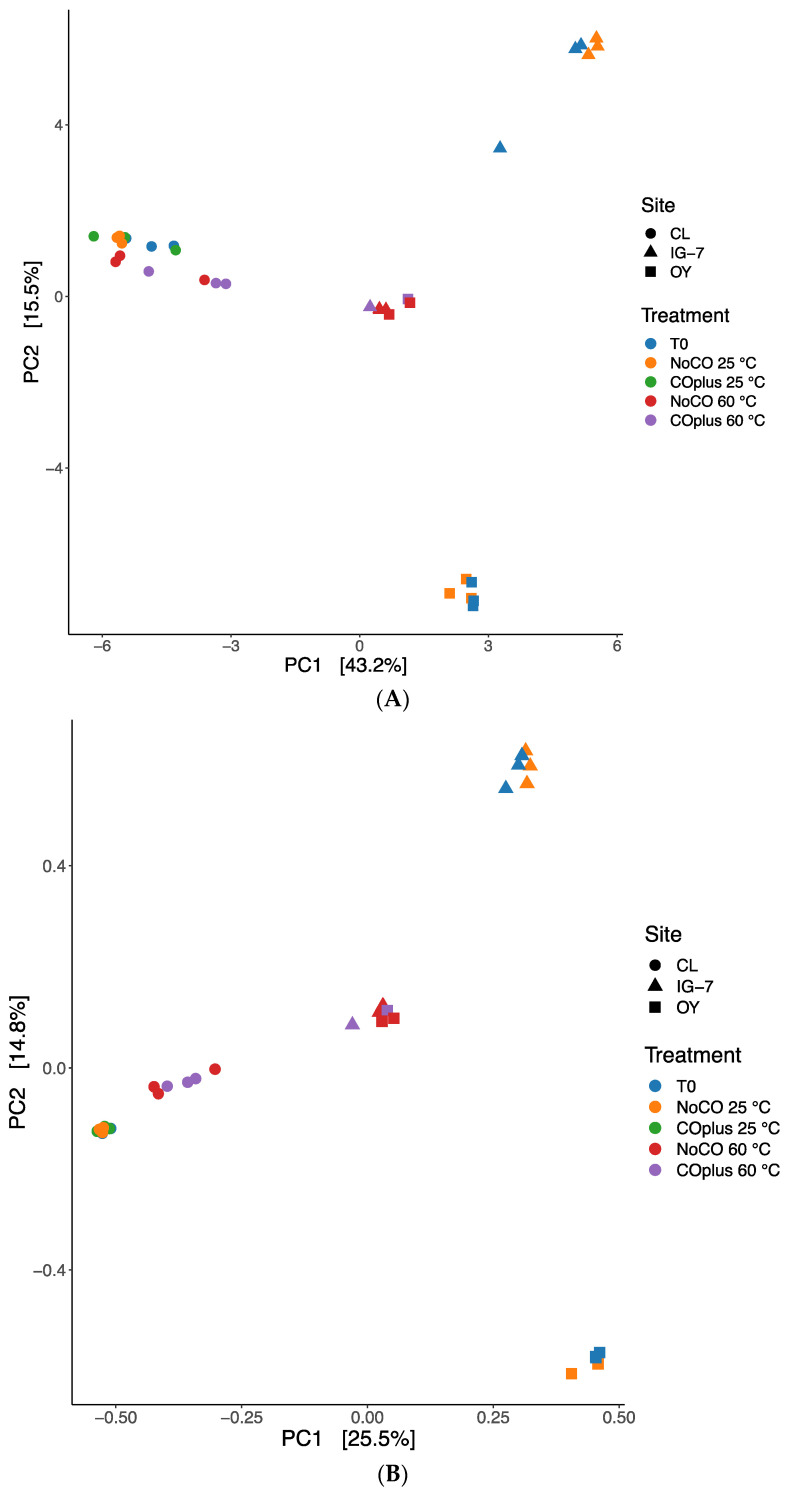
Redundancy analysis for sites CL, IG-7 and OY. Samples are grouped by treatment and site using an Aitchison distance metric (CLR transformed data, **A**) or Hellinger distance metric (**B**).

**Figure 7 microorganisms-09-00012-f007:**
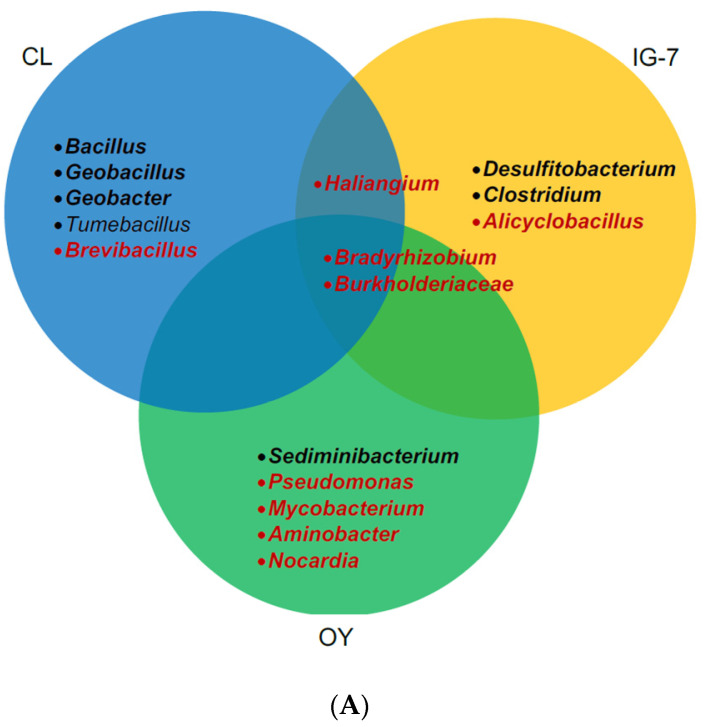
Venn diagram of the distribution of nickel- and molybdenum-dependent CO oxidizers within and among sites at 25 °C (**A**) and 60 °C (**B**); bold black italics represent known nickel-dependent CO-oxidizing genera; bold red italics represent known molybdenum-dependent CO-oxidizing genera; black italics represent putative nickel-dependent CO-oxidizing genera.

**Table 1 microorganisms-09-00012-t001:** Summary of CO uptake rates (µmol gdw^−1^ d^−1^) at CL, IG-7, and OY for 10 ppm CO under aerobic and anaerobic conditions, and 25% CO at 25 °C and 60 °C.

Site	Age (Year)	Vegetation	pH	OM%	10 ppm CO		25% CO	
					Aerobic	Anaerobic	25 °C	60 °C
**Trial 1**								
CL	ca. 800	Forest	5.2 ± 0.05	21.8 ± 0.5	0.8 ± 0.1	0.8 ± 0.1	24.5 ± 1.5	154.5 ± 6.0
IG-7	18	Grass	4.5 ± 0.03	3.1 ± 0.4	0.1± 0.02	0.1 ± 0.01	4.0 ± 2.0	31.7 ± 10.7
OY	18	Mixed, sparse	4.8 ± 0.05	1.7 ± 0.1	0.07 ± 0.01	0.04 ± 0.01	0	5.8 ± 0.8
**Trial 2**								
CL	--	--	--	--	NM	NM	20.3 ± 4.8	36.1 ± 0.6
IG-7	--	--	--	--	NM	NM	0	6.1 ± 6.1
OY	--	--	--	--	NM	NM	0	3.5 ± 3.5

Age, vegetation status, pH and organic matter (OM) data for each site are included. Rate data are included from Trial 1 and Trial 2. All values are means ± 1 standard error. NM indicates data were not measured. Vegetation, pH and OM% were the same for Trial 1 and Trial 2.

**Table 2 microorganisms-09-00012-t002:** CO uptake rates (µmol gdw^−1^ d^−1^) at sites OY (25 °C and 60 °C) and IG-7 (25 °C) for CO concentrations of 0.1%, 1%, 5%, 15%, and 25%.

Site	CO Uptake Rate (µmol gdw^−1^ d^−1^)				
	0.1% CO	1% CO	5% CO	15% CO	25% CO
OY 25 °C	0.07 ± 0.002	ND	ND	ND	ND
OY 60 °C	0.19 ± 0.01	0.37 ± 0.02	ND	ND	ND
IG-7 25 °C	0.21 ± 0.06	0.34 ± 0.01	ND	ND	ND

All values are means ± 1 standard error. ND indicates CO uptake was not detected.

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
