# Peer review of "Anaerobic Carbon Monoxide Uptake by Microbial Communities in Volcanic Deposits at Different Stages of Successional Development on O-yama Volcano, Miyake-jima, Japan"

_microorganisms, 2020, doi:10.3390/microorganisms9010012_

Round 1
Reviewer 1 Report
This work provides thorough, detailed experimental studies on the microbial ecology of CO utilizing bacteria in the microbial populations of volcanic soils from a Japanese and an Hawaiian volcanic site. Considering that it provides evidence for the presence of anaerobic mesophilic CO-oxidisers in predominantly aerobic soils, it is ground-breaking research.
In general, the manuscript is logically ordered and well written. A few suggested corrections will be listed at the end of the review. One oft-occurring language problems is the referral to 25% (or 10%) v/v CO concentrations. These of course refer to initial concentrations as readers familiar with gas fermenting bacteria will understand right away. However to serve a general audience it would be helpful to include some "initial" inserts, for example in the Abstract, line 19.
The Introduction provides a generally well documented and accurate summary of the condition of the field of carboxidotrophs and carboxydovores. One minor correction would be to move the citation of ref 14 from line 59 to line 57, after Firmicutes. This sentence which refers to SIP probing of a CO utilizing population lacks a reference (probably Brady et al,PMID: 26388850, ref 47). The general statement that little is known about Ni-COX on line 65 is justified and supports the main thesis of the work and the choice of the three hypotheses.
The Material and Methods section was generally comprehensive as expected given the many studies carried out by these authors in these areas. I was a little surprised to see no abiotic controls and maybe this is based on the experience of the authors on uptake assays like these.
Line 144 and 145 concerning the 16S sequencing also need some identification of the primers used and whether they are efficient to amplify archaeal 16S. I see only one instance in the Supplementary data (Table S3) to significant Thaumarchaeota and wondered if the paucity of Archaeal hits is a methodological artifact or could be a real effect in these mesophilic populations under CO selection.
Results
I found it surprising that CO uptake occurred with no "observable" lag (see line 175). This statement seems at variance with results reported on the next page as follows:"Apparent lag times for 0.1% CO concentrations were consistently < 1 day, while 217 apparent lags for 1% concentrations varied from 2.1 - 9.9 days."
This seems more likely. At any rate these statements need to be reconciled.
The figures speak of excellent experimental design and replicate management, however the font sizes are very tiny and must be adjusted.
The result description in lines 199-205 is confusing, maybe because of the use of "consistent". Line 201. It would be easier to interpret if the sentence turned around e.g. "the results shown for 25% were less reproducible then CO concentrations less than 25%".
Discussion
The statement that the ecological significance of Ni-COX is largely unknown is well founded. The discovery that there are significant (probably) remnants of thermophilic anaerobic CO oxidizers in these cool volcanic soils is a significant finding and confirms the status of CO utilization as a very frequent "go to" metabolism in microbial populations in nutrient depleted conditions. The discovery of new genera with possible CO oxidation potential is very interesting, however this study doesn't confirm whether the strains of Brevibacillus and Tuberibacillus are themselves COX enabled or just along for the ride during enrichment. The same applies to the Thaumarchaeota mentioned in Supplementary table S3. Overall the work confirms that CO metabolism resides in highly mobile laterally transferred genomic modules as has been mentioned in papers like reference 12, as well as the comparative description of three genomes of Caldanaerobacter with variable COX potential, see Saint Anna et al BMC Genomics 2015 Oct 7;16:757.
Author Response
We thank the reviewer for their time and input and appreciate comments that have improved the manuscript.
General:
We agree and have interested “initial” as suggested for CO concentrations.
Introduction:
The citation has been moved as suggested and citation [47] has been inserted for the comment about SIP.
Methods:
Thanks for pointing out the missing primers, we have added them: 515f and 806r. They aren’t the best for Archaea, but in other studies we have certainly had no problem picking up Crens, Eurys, Thaums and others.
Results:
We have used killed (autoclaved) controls extensively in previous work and have yet to see any indication of abiological CO uptake//oxidation.
The reviewer is correct. We saw no “observable” lag times at 10 ppm. Uptake was apparent within the few hours that elapsed from CO addition to the first sampling point. At higher concentrations, e.g., 1000 ppm (0.1%) and up, we observed a lag before significant decreases in CO were detected. The lags were actually consistent with changes that might be anticipated from the rates at 10 ppm. As we stated in the results and discussion, we view these as “apparent” lags, not as periods of no activity. They reflect the limitations of the analytical system, not what the bacteria were doing.
We agree, there was some confusion about the meaning of consistency; we have adopted the suggestion of “reproducibility” instead. We also included a specific statement about the numbers of replicates that were positive in the two trials.
We appreciate the reviewer’s comments and have increased the font sizes. They actually started out at 12-14 point, but when inserted in the text rather than presented on separate pages, the sizes appeared too small.
Discussion:
We agree, there is much more to learn about Ni-dependent CO oxidizers. We were able to observe changes in some populations, but only suggest that these taxa are putative CO oxidizers. It’s possible that they grow along with primary CO oxidizers, however, we would note that given the low cell yields that occur with CO as we stated in the discussion, there doesn’t seem to be a lot of scope for a second population to grow using carbon from another. Resolution of the responding populations will likely new approaches targeting the cooS gene.
Reviewer 2 Report
Summary
The authors achieved a comparative analysis of CO uptake of different samples taken from 3 different sites from O-yama Volcano on Miyake-jima Island (Japan), at different temperatures (25ºC and 60ºC) and presence/absence of CO. The 3 sites correspond with a forest of 800-yr old volcanic deposit and two additional sites impacted by a recent eruption 18 years ago with high or low plant colonization. They determine the phyla and the genera present in each site by 16S rRNA analysis obtaining the microbial community. Finally, they conclude that CO uptake of thermophiles is higher than the mesophilic, and the CO uptake rates increase with organic matter in volcanic deposits. The authors speculate about the reasons of the microbial communities responses to different variables and propose a potential novel Ni-COX diversity.
Major comments.
- Table 1: in this table, the authors show the CO uptake rates at sites CL, IG-7 and OY. However, the table show a second set of data under the title “16S rRNA analysis” that it is not explained in the foot of the table. Moreover, the mention to the 16S rRNA analysis appears in the Introduction and in the Discussion, but it does not appear any analysis in the Results. They should explain the meaning of these results in this section.
- Regarding to the data of the Figure 1 and Table 1, the authors mention in Lines 199-200 that “CO uptake for 25% was less consistent for IG-7 and OY replicates”. However, attending to the data, I just see less consistent the data of IG-7 at 60ºC. IG-7 at 25ºC and OY at 60ºC show a small standard deviation, so I don’t understand the previous sentence. Could you clarify this point?
- Following with the previous point, in the Table 2 the authors show a second set of data with a screening of CO concentrations trying to obtain data that are more consistent. However, the CO uptake rate is not detected in any case at 25% of CO. In the Discussion (Lines 378-387), the authors suggest that Ni-COX populations could be inhibited by high concentrations of CO. So, how do you explain this difference with the data of Table 1 at 25% of CO?
- Lines 211-217. The authors explain the results of an experiment where they measure the lag time of incubations in the different samples. It would be desirable to represent these data in a graphic.
- In the Introduction and the Discussion, the authors mention and analyze the ecological significance of Ni-COX in the CO uptake. However, this enzyme is not mentioned in the Results section. Taking into account the deep analysis achieved during the Discussion, it would be very useful to point out, for instance in the Figure 2, which genera present CO dehydrogenases dependent of Ni or Mo. In fact, it could be shown this information in a graphic where it was possible to compare the presence of Ni-COX between the different sites, the temperatures 25ºC and 60ºC, and the presence or absence of CO.
Minor comments.
- Figure 1: All the text and numbers included in the graphic are extremely small so I recommend increasing their sizes.
- I extend the previous comment to the most of the graphics of the paper.
- Line 200: Change “the results…” by “The results…”
Author Response
We appreciate the reviewer's comments. They have improved the manuscript. We have incorporated suggestions as indicated below.
Major comments:
Comment 1. The table caption has amended to more clearly reference the two distinct trials that were used for measuring CO uptake rates; these are also described more clearly throughout the text (introduction, methods, results and discussion). Specifically, the first trial included activity at 10 ppm and at 25% CO; the second trial only used 25% CO and the soils in this trial were sub-sampled for use in 16S rRNA gene sequence analyses.
Comment 2. Consistency referred to the fact that at sites IG-7 and OY, the number of replicates that was active for each treatment varied between the trials. Text was added to increase clarity about this point (line 192).
Comment 3. We agree that this is a puzzling behavior. It is also unusual in our experience with assays at low CO concentrations and with assays involving non-toxic gases, e.g., methane. However, a survey of a much larger set of sites has revealed a similar behavior. Thus, the variability in response at 25% appears to be an inherent property of soil microbial communities. It reflects either a stochastic element of the response or subtle changes in communities from one point to the next. Notably, this is a phenomenon at high CO concentrations, not at lower levels.
Comment 4. We agree with the reviewer and have added a new figure that shows the lag time results.
Comment 5. We use Ni-COX to refer to nickel-dependent CO oxidizers and Ni-CODH to refer to the nickel-dependent CO dehydrogenase as defined in the text. We understand the reviewer’s point though. We have created a new figure in the form of a Venn diagram that illustrates the known Ni-COX and Mo-COX and suspected Ni-COX genera for each site at 25ËšC and 60ËšC. More detailed information is also provided in the text.
Minor comments:
Comment 1 and 2. We agree and have increased font sizes as much as we can for the various figures.
Comment 3. Corrected as suggested.
Reviewer 3 Report
The manuscript under review deals with the study of CO oxidation in anaerobic microbial communities of different soils under mature forest (more than 800 years old), and relatively young soils covered by grass after relatively recent volcanic eruption (year 2000) in Japan. Authors selected different modes of incubations, varying with temperatures (25 and 60oC) and CO concentrations (from 10 ppm to 25%). Accordingly, genomic analysis of correspondent microbial communities based on 16S rRNA sequences were examined, and that reveals a potential new representatives of anaerobic CO oxidizers from genera of Brevibacillus and Tuberibacillus.
The manuscript is well written and might be published after a minor revision.
Some specific questions to be answered:
- Why authors choose the temperature of 60oC for soil samples incubation? The native soils newer reach this high temperature!
- Why authors not isolated this new potential anaerobic CO oxidizers of Brevibacillus and Tuberibacillus genera they detected in the communities? For journal like “Microorganisms” I think it is an obligation to isolate the new microbes to be published.
- Table 1: units for 16S rRNA analysis are not included in table.
- Latin name of all taxa must be italysed including manuscript text and lit. list (lines 509, 609, 634, 637 to note a few).
Author Response
We thank the reviewer for comments and input. We have used them to improve the manuscript as indicated below.
Comment 1. It’s true that few soils see temperatures of 60ËšC. Nonetheless, the presence of thermophiles in soils is well, even in those that rarely, if ever, rise about 30ËšC. As a result, one can expect to find thermophilic processes if one elevates temperature into a thermophilic range. Our goal was to do just that, because as we stated in the introduction and discussion, Ni-COX are primarily known as thermophilic. Our results show for the first time that thermophilic and mesophilic Ni-COX co-occur and provide new insights about the ecological distributions of both.
Comment 2. We agree that isolation is important. We have isolated numerous molybdenum-dependent CO oxidizers in our previous studies. However, isolation is a separate task that is beyond the scope of the work reported here. Descriptions of isolates also were not required for the special issue of Microorganisms to which this manuscript has been submitted.
Comment 3. Table 1 describes rates of CO uptake. The column header might have caused some confusion, so we have relabeled the headers for clarity.
Comment 4. Thank you for pointing this out. We simply missed some of these in our editing. They have all be italicized now.
Round 2
Reviewer 2 Report
The authors have achieved all the suggested changes so I don't have anymore major comments.
Just to point out that they have increased the size of the graphics and I think that it is not necessary. I proposed to increase just the text and the numbers of the graphics.
Author Response
Thank you for your comments.